# Reduced Replication of Highly Pathogenic Avian Influenza Virus in Duck Endothelial Cells Compared to Chicken Endothelial Cells Is Associated with Stronger Antiviral Responses

**DOI:** 10.3390/v14010165

**Published:** 2022-01-17

**Authors:** Anja C. M. de Bruin, Monique I. Spronken, Theo M. Bestebroer, Ron A. M. Fouchier, Mathilde Richard

**Affiliations:** Department of Viroscience, Erasmus Medical Center, 3015 GD Rotterdam, The Netherlands; a.c.m.debruin@erasmusmc.nl (A.C.M.d.B.); m.spronken@erasmusmc.nl (M.I.S.); t.bestebroer@erasmusmc.nl (T.M.B.); r.fouchier@erasmusmc.nl (R.A.M.F.)

**Keywords:** influenza A virus, highly pathogenic avian influenza, endothelial cells, chicken, duck, cell tropism, pathogenesis, innate immunity, primary cell culture

## Abstract

Highly pathogenic avian influenza viruses (HPAIVs) cause fatal systemic infections in chickens, which are associated with endotheliotropism. HPAIV infections in wild birds are generally milder and not endotheliotropic. Here, we aimed to elucidate the species-specific endotheliotropism of HPAIVs using primary chicken and duck aortic endothelial cells (chAEC and dAEC respectively). Viral replication kinetics and host responses were assessed in chAEC and dAEC upon inoculation with HPAIV H5N1 and compared to embryonic fibroblasts. Although dAEC were susceptible to HPAIV upon inoculation at high multiplicity of infection, HPAIV replicated to lower levels in dAEC than chAEC during multi-cycle replication. The susceptibility of duck embryonic endothelial cells to HPAIV was confirmed in embryos. Innate immune responses upon HPAIV inoculation differed between chAEC, dAEC, and embryonic fibroblasts. Expression of the pro-inflammatory cytokine *IL8* increased in chicken cells but decreased in dAEC. Contrastingly, the induction of antiviral responses was stronger in dAEC than in chAEC, and chicken and duck fibroblasts. Taken together, these data demonstrate that although duck endothelial cells are permissive to HPAIV infection, they display markedly different innate immune responses than chAEC and embryonic fibroblasts. These differences may contribute to the species-dependent differences in endotheliotropism and consequently HPAIV pathogenesis.

## 1. Introduction

Avian influenza A viruses (AIVs) are maintained through enzootic circulation in wild waterfowl, predominantly in the orders of the *Anseriformes* (e.g., ducks, geese, and swans) and *Charadriiformes* (e.g., gulls) [1]. Influenza A viruses are classified based on the antigenic properties of their surface glycoproteins hemagglutinin (HA) and neuraminidase (NA). To date, sixteen HA subtypes (H1–H16) and nine NA subtypes (N1–N9) have been identified in wild waterfowl [2]. Infections with influenza viruses in wild waterfowl are mostly asymptomatic and do not cause histological lesions [3,4]. In these species, AIV tropism is limited to the digestive tract [5]. As a result, the route of transmission among wild waterfowl is thought to be primarily fecal–oral. Occasionally, AIVs spillover from wild birds to terrestrial poultry, e.g., chickens and turkeys, via (in)direct contact [6]. In terrestrial poultry, AIVs can be of low pathogenicity causing mild or subclinical disease in the respiratory and gastrointestinal tracts [7]. These viruses were coined low-pathogenic avian influenza viruses (LPAIVs). However, viruses of the H5 and H7 subtypes can mutate into highly pathogenic avian influenza viruses (HPAIVs) in terrestrial poultry, leading to severe systemic infections with mortality rates reaching 100% [7].

The main determinant for the high virulence of HPAIVs in terrestrial poultry is related to the proteolytic cleavage site in the HA protein [8]. The HA protein mediates binding of virus particles to sialic acid-containing cell surface receptors and, after endocytosis, fusion of the viral and endosomal membranes. HA is synthesized as HA0 precursor and requires post-translational cleavage into two chains, HA1 and HA2, in order to become fusogenic upon acidic pH [9,10]. LPAIVs possess a monobasic cleavage site, consisting of one basic amino acid (arginine or lysine), which is cleaved by host trypsin-like proteases [11]. Such proteases are locally expressed in the respiratory and gastrointestinal tracts, explaining the LPAIV tropism in terrestrial poultry. In contrast, the cleavage site of HPAIVs contain multiple basic amino acids. This multibasic cleavage site (MBCS) is cleaved by ubiquitously expressed proprotein convertases, such as furin and PC5/6 [12,13]. This allows for replication of HPAIVs in a plethora of different cell types and tissues, which underlies the systemic spread of HPAIVs in terrestrial poultry [14].

An important hallmark of pathogenesis in HPAIV-infected terrestrial poultry is the infection of endothelial cells. This endotheliotropism is observed in many different organs, including the lung, heart, liver, and brain [15,16,17,18,19,20], and is dependent on the presence of an MBCS [21,22,23]. Endothelial cells form the barrier between the cardiovascular system and surrounding tissues, but they also maintain vascular homeostasis, influence coagulation processes, and orchestrate immune responses [24]. Therefore, viral infections that result in the dysfunction or loss of endothelial cells often lead to severe systemic complications which present with hemorrhage, edema, and coagulopathy [25]. Likewise, infection and general dysfunction of endothelium in HPAIV-infected chickens results in hemorrhaging and edema [17,19,26] and is related to HPAIV-induced disseminated intravascular coagulation [27], impaired thermoregulation [28], and profuse inflammatory cell recruitment [26]. Furthermore, excessive cytokine responses have been reported in HPAIV-infected chickens [29,30], which are reminiscent of endothelium-driven pro-inflammatory responses in mammals [31]. HPAIV infections in terrestrial poultry eventually lead to fatal damage to the cardiovascular system and/or multi-organ failure [7].

HPAIVs are associated with outbreaks in poultry populations and not in wild birds. The major exceptions are HPAIVs of the H5 subtype belonging to the A/goose/Guangdong/1/96-lineage (GsGd-H5). GsGd-H5 HPAIVs have progressively spread to multiple countries on four continents since 2003, repeatedly causing outbreaks in both wild bird and poultry populations [32]. Sporadically, non-GsGd-H5 HPAIVs are detected in wild birds [33,34,35,36,37]. Clinical manifestations upon infection of wild birds with HPAIVs are in stark contrast with those in terrestrial poultry [7]. Experimental inoculation of non-GsGd-H5 HPAIVs generally does not result in overt disease in mallards or domestic ducks [38,39,40,41,42]. The pathogenicity of GsGd-H5 HPAIVs upon natural or experimental infection is variable, ranging from asymptomatic infection to fatal neurological implications [7,43,44], and is dependent on the virus strain, bird species, and host age [45]. Regardless of clinical signs, HPAIV-inoculated ducks are generally productively infected, can shed and subsequently transmit virus efficiently [38].

In contrast to terrestrial poultry, HPAIV are not endotheliotropic in wild birds regardless of disease severity, and virus antigen is mostly found in the parenchyma, not in the vasculature [46]. Exceptions to this rule are certain species of swan [47] and ducks infected with GsGd-H5 HPAIVs from 2016 [48,49]. The stark differences in endotheliotropism of HPAIVs in terrestrial poultry and wild birds raise the question as to how and why endothelial cells of wild birds are resistant to HPAIV infection in vivo. Studies on this topic are limited, due to the recent application of robust primary endothelial cell culture methods to avian species other than chicken [50,51,52,53,54,55]. We have developed methods to culture primary duck endothelial cells from embryonic aortic arches and bone marrow [56]. Here, we compared virus replication in and host responses of primary chicken and duck aortic endothelial cells (AEC) upon inoculation with HPAIV in order to elucidate the species-specific endotheliotropism and how innate immune responses might influence HPAIV-induced immunopathology.

## 2. Material and Methods

### 2.1. Cell Culture

Primary endothelial cells were cultured from aortic arches of chicken, *Gallus gallus domesticus,* and Pekin duck, *Anas platyrhynchos domesticus*, embryos as described previously [54,56]. Eighteen-day-old chicken embryos were obtained from Drost Loosdrecht B.V., The Netherlands, and 21-day-old duck embryos were obtained from Duck-To-Farm B.V., The Netherlands. In the European Union, embryos from avian species are not subjected to ethical regulations, regardless of the embryonic stage. Briefly, ascending aortic arches were separated from the hearts and minced into small pieces. The pieces were transferred to culture dishes coated with 0.2% gelatin (Sigma-Aldrich, St. Louis, MO, USA) and maintained in Microvascular Endothelial Cell Growth Medium-2 (EGM^TM^-2MV; LONZA, Basel, Switzerland). Endothelial cells were passaged every 3–4 days for a minimum of 15 passages before use in infection experiments. Chicken and duck embryonic fibroblasts (CEF; DEF) were isolated from 11-day-old chicken and 13-day-old duck embryos (protocol adapted from [57]) and cultured in Medium 199 (LONZA, Basel, Switzerland) supplemented with 10% fetal calf serum (FCS; Greiner Bio-One, Kremsmünster, Austria), 10% Tryptose Phosphate Broth (TPB; MP Biomedicals, Santa Ana, CA, USA), 100 U/mL penicillin (LONZA, Basel, Switzerland), and 100 U/mL streptomycin (LONZA, Basel, Switzerland).

Madin–Darby canine kidney cells (MDCK) were cultured in Eagle’s Minimal Essential Medium (LONZA, Basel, Switzerland) supplemented with 10% FCS, 100 U/mL penicillin, 100 U/mL streptomycin, 2 mM L-glutamine (LONZA, Basel, Switzerland), 1.5 mg/mL sodium bicarbonate (LONZA, Basel, Switzerland), 20 mM HEPES (LONZA, Basel, Switzerland), and non-essential amino acids (NEAA; LONZA, Basel, Switzerland). Human Embryonic Kidney-293T (HEK-293T) cells were cultured in Dulbecco’s Modified Eagle Medium (LONZA, Basel, Switzerland) supplemented with 10% FCS, 100 U/mL penicillin, 100 U/mL streptomycin, 2 mM L-glutamine, 1 mM sodium pyruvate (Gibco, Waltham, MA, USA), 0.5 mg/mL geneticin (Invitrogen, Waltham, MA, USA), and NEAA. EA-hy926 (human umbilical vein endothelial cells) and NCl-H441 (human papillary adenocarcinoma lung epithelial cells) cells were obtained from the ATCC (CRL-2922 and HTB-174, respectively) and maintained in Roswell Park Memorial Institute medium (LONZA, Basel, Switzerland) supplemented with 10% FCS, 100 U/mL penicillin, and 100 U/mL streptomycin. Pooled primary human umbilical vein endothelial cells (HUVEC) were obtained from PromoCell (C-12203; Heidelberg, Germany) and cultured in EGM^TM^-2MV on gelatin-coated (0.2%) plastic ware. Cells of mammalian origin were incubated at 37 °C in 5% CO_2_, and avian cells were cultured at 40 °C in 5% CO_2_.

### 2.2. Virus Origin and Propagation

Wild-type HPAIV isolate A/Vietnam/1203/04 (H5N1) was kindly provided by Robert Webster (St. Jude Children’s Research Hospital, Memphis, TN, USA) and propagated in 11-day-old embryonated chicken eggs (ECE) (passage xE3). HPAIV A/turkey/Turkey/1/05 (H5N1), containing all 8 segments of A/turkey/Turkey/1/05, was rescued by reverse genetics (RG) [58] and propagated in MDCK cells (2 passages). Infectious virus titers were determined by end-point titration in MDCK cells and expressed as median tissue culture infectious dose (TCID_50_/mL), as previously described [59]. All HPAIV experiments were performed in biosafety level 3 containment facilities at the Erasmus Medical Center, Rotterdam, The Netherlands.

### 2.3. Tube Formation Assay

The capacity of cells to form vasculature-like structures was assessed by tube formation assays. HUVEC, MDCK, chAEC, or dAEC were trypsinized, and 1.5 × 10^4^ cells were seeded in 100 µL EGM^TM^-2MV, per 96-well plate well, on top of a layer (50 µL) of solidified growth factor deprived basal membrane extract Matrigel^®^ (Corning, Glendale, AZ, USA). Plates were incubated for 4 h at 37 °C in 5% CO_2_ (HUVEC and MDCK) or 40 °C in 5% CO_2_ (chAEC and dAEC) and imaged using the Axio Vert.A1 inverted microscope (Zeiss, Oberkochen, Germany).

### 2.4. Acetylated Low-Density Lipoproteins Uptake

EA-hy926, NCl-H441, chAEC, and dAEC were incubated for 4 h (EA-hy926 and NCl-H441 at 37 °C in 5% CO_2_; chAEC and dAEC at 40 °C in 5% CO_2_) in complete culture medium containing 3.3 μg/mL Alexa Fluor^®^488-conjugated acetylated low-density lipoproteins (ac-LDL; Thermo Fisher Scientific Scientific, Waltham, MA, USA). Cells were washed and visualized using a Laser Scanning Microscope 700 (Zeiss, Oberkochen, Germany) and flow cytometry (FACSLyric, BD Biosciences, Franklin Lakes, NJ, USA). Data were analyzed using FlowJo v10.7.2. software (BD Biosciences, Ashland, OR, USA).

### 2.5. Lectin Staining

The presence of α2,3/α2,6-linked sialic acids on HEK-293T cells, chAEC, and dAEC was assessed by flow cytometry. Confluent monolayers were treated for 2 h at 37 °C in 5% CO_2_ with fresh EGM^TM^-2MV with or without 100 mU/mL neuraminidase from *Vibrio cholera* (Sigma-Aldrich, St. Louis, MO, USA), which enzymatically removes sialic acid moieties to act as negative control. Cells were washed, trypsinized, and incubated with PBS-2%FCS containing 10 µg/mL biotinylated *Maackia amurensis* lectin II (MAL-II; Vectorlabs, Burlingame, CA, USA; α2,3-sialic acid specific) or 10 µg/mL biotinylated *Sambucus nigra agglutinin* lectin (SNA; EY labs, San Mateo, CA, USA; α2,6-sialic acid specific) for 30 min at 4 °C. Biotin-labeled cells were stained with 5 μg/mL FITC-conjugated streptavidin (F0422; Agilent, Santa Clara, CA, USA) for 30 min at 4 °C, detected by flow cytometry, and analyzed using FlowJo v10.7.2 software (BD Biosciences, Ashland, OR, USA).

### 2.6. Detection of the Nucleoprotein by Flow Cytometry

To test the susceptibility of MDCK, CEF, DEF, chAEC, and dAEC to HPAIV infection, confluent monolayers were inoculated with H5N1 A/Vietnam/1203/04 at a multiplicity of infection (MOI) of 1 (TCID_50_/cell) in corresponding culture media without FCS and TPB. Cultures were incubated at 40 °C in 5% CO_2_. At 6 h post inoculation (hpi), cells were washed and trypsinized for detection of nucleoprotein (NP) as a marker for infection. Cells were fixed and permeabilized with BD Cytofix/Cytoperm™ (BD Biosciences, Franklin Lakes, NJ, USA) according to manufacturer’s instructions and subsequently stained for 30 min at 4 °C with 20 µg/mL anti-influenza NP antibody (H16-L10-4R5 (ATCC^®^ HB-65™)) diluted in BD Perm/Wash™ buffer, followed by 10 µg/mL Alexa Fluor^®^488-conjugated goat anti-mouse IgG2α antibody (A-21131; Thermo Fisher Scientific, Waltham, MA, USA). Cells were subjected to flow cytometry, and percentage of NP-positive cells was analyzed using FlowJo v10.7.2 software (BD Biosciences, Ashland, OR, USA).

### 2.7. Viral Replication Kinetics

Confluent monolayers of CEF, DEF, chAEC, or dAEC were inoculated with H5N1 A/Vietnam/1203/04 at an MOI of 1 or 0.001. After 1 h of incubation, the inoculum was removed, and the cells were washed thrice with PBS. Fresh serum-free medium (M199 for EF and EGM^TM^-2MV for AEC) was overlaid, and cultures were incubated at 40 °C in 5% CO_2_. Viral replication curves were generated in the absence of exogenous trypsin. Supernatant was harvested at the specified time points and stored at −80 °C until further analysis. Infectious virus titers in the supernatant were determined by end-point titration in MDCK cells, and viral matrix segment copy number was determined by reverse transcriptase quantitative PCR (RT-qPCR) (see Section 2.8).

### 2.8. RT-qPCR

To determine innate immune gene expression patterns, monolayers of CEF, DEF, chAEC, and dAEC were inoculated with H5N1 A/Vietnam/1203/04 at an MOI of 1 in serum-free medium (M199 for EF and EGM^TM^-2MV for AEC). After 1 h of incubation, the inoculum was removed, and the cells were washed thrice with PBS. Fresh serum-free medium was overlaid, and cultures were incubated at 40 °C in 5% CO_2_. Mock controls were treated with medium only, since the percentage of allantoic fluid in the virus-containing inoculum was limited (<1%). At 6 hpi and 12 hpi, total RNA from virus- and mock-inoculated cells was extracted using the High Pure RNA Isolation Kit (Roche, Basel, Switzerland) according to manufacturer’s instructions. RNA was extracted similarly from 200 µL replication curve supernatant (see Section 2.7). Concentration and quality of the RNA was determined using a NanoDrop™ spectrophotometer (Thermo Fisher Scientific, Waltham, MA, USA). For cDNA synthesis, 100 ng of RNA was reverse-transcribed using oligo (dT) primers (Thermo Fisher Scientific, Waltham, MA, USA) and SuperScript^®^ IV Reverse Transcriptase (Thermo Fisher Scientific, Waltham, MA, USA) according to manufacturer’s instructions. Gene expression levels were assessed by dye-based qPCR using the primer pairs listed in Table 1, targeting *GAPDH* (glyceraldehyde-3-fosphate dehydrogenase), *IL6* (interleukin-6), *IL8* (interleukin-8), or *RSAD2* (viperin), and SYBR^®^ Green PCR Master Mix (Applied Biosystems, Waltham, MA, USA). Amplification and detection were performed on an ABI7700 (Thermo Fisher Scientific, Waltham, MA, USA) according to manufacturer’s instructions. Alternatively, 5 µL of RNA was directly added to a mix for RT-qPCR, containing the primer and probes listed in Table 1, targeting the influenza matrix (M) segment, pan-species *GAPDH*, or chicken or duck *IFNB* (interferon-beta) and 4X TaqMan™ Fast Virus 1-Step Master Mix (Thermo Fisher Scientific, Waltham, MA, USA). M segment copy number was quantified using a standard curve performed with the VetMAX^TM^ AIV control kit (Thermo Fisher Scientific, Waltham, MA, USA). The following cycling program was used on an ABI7700: 5 min 50 °C, 20 s 95 °C, (3 s 95 °C, (duck *IFNB*: +30 s 54 °C), 31 s 60 °C) × 45 cycles. PCR efficiency, linear range, and sensitivity were determined for all primer sets using cDNA or PCR products. Fold changes were calculated using the 2^−ΔΔCT^ method with *GAPDH* serving as a housekeeping gene for normalization and mean mock values as baseline reference.

### 2.9. Detection of Von Willebrand Factor mRNA by RT-PCR

Total RNA was isolated from chAEC and dAEC as described in Section 2.8. For cDNA synthesis, 100 ng of RNA was reverse-transcribed using random primers (Promega, Madison, WI, USA) and SuperScript^®^ IV RT. Expression of von Willebrand Factor (vWF) mRNA in dAEC was determined by PCR as described before [56]. VWF mRNA expression in chAEC was determined using primers listed in Table 1 and the AmpliTaq Gold^TM^ DNA Polymerase kit (Thermo Fisher Scientific, Waltham, MA, USA) according to manufacturer’s instructions. The following cycling program was used: 6 min 95 °C, (30 s 95 °C, 1 min 55 °C, 20 s 72 °C) × 30 cycles. PCR product sizes were analyzed by gel electrophoresis, and the amplicons were Sanger sequenced to confirm PCR specificity.

### 2.10. In Ovo Inoculation and Immunohistochemical Analysis

ECE (14- and 17-day-old) or embryonated duck eggs (EDE; 18- and 21-day-old) were inoculated with 10^3^ TCID_50_ of HPAIV RG-A/turkey/Turkey/1/05 via the allantoic route. Eggs were incubated in a humidified chamber at 37 °C for 24 h (14d ECE; 18d EDE) or 48 h (17d ECE; 21d EDE), after which they were candled and chilled for a minimum of 1 h. The chorio-allantoic membrane (CAM) was harvested, followed by decapitation of the embryo and immersion of all tissues in 10% neutral buffered formalin. After 2 weeks of fixation, tissues were decalcified for 4 days in 10% EDTA and embedded in paraffin. Thin (3 µm) sections were prepared for a hematoxylin and eosin (HE) staining and immunohistochemical (IHC) analysis. Formalin-fixed, paraffin-embedded (FFPE) sections were rehydrated, and antigens were retrieved by treatment with 0.1% protease XIV from *Streptomyces griseus* (Sigma-Aldrich, St. Louis, MO, USA) in PBS for 10 min at 37 °C. Endogenous peroxidase activity was blocked by treatment with 3% H_2_O_2_ in PBS for 10 min at room temperature. Sections were incubated with 5 μg/mL anti-NP antibody or isotype control (mouse IgG2α; MAB003; R&D Systems, Minneapolis, MN, USA) for 1 h at room temperature, followed by 1 h incubation with 10 µg/mL detection antibody (HRP-coupled goat anti-mouse IgG2α (Bio-Rad, Hercules, CA, USA; Star133P)). Subsequently, sections were developed with 3-amino-9-ethyl-carbazole (Sigma-Aldrich, St. Louis, MO, USA) in N,N-dimethylformamide (Honeywell Fluka, Charlotte, NC, USA) diluted in a final concentration of 0.0475 M of sodium acetate (NaAc, pH = 5) with 0.05% of H_2_O_2_ for 10 min at room temperature and counterstained with hematoxylin. Glass coverslips were mounted using Kaiser’s glycerol gelatin (VWR, Radnor, PA, USA). Pictures were taken on the Microscope Axio Imager.A2 (Zeiss, Oberkochen, Germany). White balance was adjusted with Adobe Photoshop 2021.

### 2.11. Statistical Analysis

Statistical analyses were performed as described in the figure legends of Figures 3 and 4. Additionally, upregulation of genes, as described in the main text regarding Figure 4, was defined by a statistically significant change from baseline mRNA expression as determined by unpaired *t*-tests. Analyses were performed using Graph Pad Prism 9 (GraphPad Software Inc., San Diego, CA, USA). Data points in graphs are depicted as mean ± standard deviation (SD) and consist of three independent experiments or as otherwise stated in the figure legends. A *p*-value < 0.05 was considered to be significant.

## 3. Results

### 3.1. Primary Avian Aortic Endothelial Cells Have Endothelial Cell Characteristics

Embryonic endothelial cells were isolated from the ascending aortas of 18-day-old ECE and 21-day-old EDE as described previously [54,56]. Pekin duck embryos were used for all experiments because they are the domestic equivalent of mallards and show comparable pathogenesis, viral replication levels, and resistance to HPAIV-induced morbidity and mortality as wild ducks [43]. The aortic cells were passaged in endothelial-cell specific medium containing growth factors until they showed a bona fide endothelial cell phenotype and cuboidal morphology (Appendix A). This required a minimum of 14 passages, from which point onward the cells were used for infection experiments until they reached replicative senescence after passage 25. Each preparation of AEC was tested for endothelial cell characteristics, using human endothelial cells, HUVEC or EA-hy, as positive controls. The AEC formed vascular-like structures in a tube formation assay (Figure 1A) and took up acetylated low-density lipoprotein (Ac-LDL) via their scavenger receptors (Figure 1B,C), a characteristic of endothelial cells [66]. Furthermore, the AEC expressed mRNA coding for the endothelial cell-specific von Willebrand Factor protein (Figure 1D).

### 3.2. Duck AEC Mainly Express α2,3-Linked Sialic Acid Moieties

The HA of AIVs has a preference for binding to α2,3-linked sialic acids (SA) [67,68] as opposed to human influenza A viruses, which bind preferentially to α2,6-linked SAs [69,70]. Lectin stainings were performed to determine which SA moieties are expressed on the surface of primary chAEC and dAEC. As shown in Figure 2, MAL-II lectins, which are specific for α2,3-linked SAs, strongly bound to chAEC and dAEC. The specificity of the MAL-II staining was confirmed by a reduction in signal upon sialidase treatment. SNA lectins did not bind chAEC, indicating the absence of α2,6-linked SAs, whereas dAEC showed a slight positivity for α2,6-linked SAs. HEK-293T cells were used as positive control for MAL-II and SNA lectin binding. The presence of α2,3-linked SAs and absence of α2,6-linked SAs corresponds to what has already been described for chAEC [54]. The presence of α2,3-linked SAs suggests that dAEC are eligible host cells for the initial step of the influenza A virus replication cycle, which is in accordance with our previous study in which dAEC were stained positively for NP upon inoculation with HPAIV at high MOI [56].

### 3.3. Productive Infection of HPAIV H5N1 in dAEC

Duck AEC were previously shown to be susceptible to H5N1 HPAIVs following high MOI inoculation [56]. Here, we extended the previous analysis by comparing initial infection percentages upon inoculation with HPAIV H5N1 at an MOI of 1 between chAEC, dAEC, and primary embryonic fibroblast cultures (CEF and DEF) at 6 hpi (Figure 3A). Embryonic fibroblasts were included to discern species-related differences from cell type-related differences. The GsGd-lineage H5N1 HPAIV isolate A/Vietnam/1203/04 was used as it replicates to high titers in Pekin ducks without infecting the endothelium [21]. CEF, chAEC, and dAEC showed significantly lower infection percentages than MDCK cells, a cell line highly susceptible to influenza A viruses. Duck AEC showed a trend towards lower infection percentages compared to chAEC, CEF, and DEF, albeit the differences were not statistically significant. No statistically significant differences were observed in the amount of virus produced upon the first cycle of H5N1 replication in all investigated cell types, as shown by viral RNA copy number and infectious titer in the supernatant (Figure 3B). Subsequently, to assess whether dAEC can sustain multi-cycle replication of HPAIV, replication of H5N1 HPAIV at low MOI was compared between chAEC and dAEC. H5N1 replicated to lower viral titers in dAEC than in chAEC, showing significant differences at 24 and 48 hpi (Figure 3C). This coincided with a delay in the onset of cytopathic effects in dAEC (Appendix A). The multi-cycle replication of HPAIV in dAEC confirms the expression of proteases that can cleave and activate HPAIV HA. Together, these data indicate that dAEC are not inherently resistant to HPAIV infection, but duck endothelial cells might be more potent in limiting HPAIV infection than chicken endothelial cells.

### 3.4. Differential Host Responses in dAEC upon HPAIV Inoculation Compared to chAEC and DEF

In chickens, H5N1 HPAIV infections are associated with excessive cytokine responses, which are hypothesized to contribute to the high morbidity and mortality [29,30]. Exuberant cytokine responses are infrequent in HPAIV-infected ducks, which primarily mount an antiviral response in the affected tissues (reviewed in [71,72]). The innate immune responses in AEC upon HPAIV A/Vietnam/1203/04 inoculation were determined to assess whether chicken and duck primary cultures respond differentially and whether these differences correlate with the reduced HPAIV replication in dAEC. Relative gene expression levels of a subset of immune genes were analyzed in chAEC, dAEC, CEF, and DEF at 6 and 12 hpi upon inoculation with HPAIV H5N1 at an MOI of 1. Only minor mRNA level fold changes were detected for the pro-inflammatory cytokines *IL6* and *IL8* (Figure 4A), which have previously been shown to be upregulated in HPAIV-infected chickens but rarely in ducks [73]. Gene expression levels of *IL6* were not markedly altered upon infection with H5N1 HPAIV in any cell type, but *IL8* fold change values were higher in chicken than in duck cells. Although *IL8* expression levels were unaltered in DEF, *IL8* expression significantly decreased in dAEC at 6 and 12 hpi. Contrastingly, upregulation of *IFNB*, a type-I-interferon-encoding gene that orchestrates antiviral defenses, was the highest in dAEC. *IFNB* expression was increased significantly more at 12 hpi in dAEC than in chAEC (Figure 4B). The *IFNB* upregulation in dAEC correlated with a significantly higher fold change value of the interferon stimulated gene (ISG) *RSAD2*, coding for the antiviral protein viperin, than in chAEC and in DEF (Figure 4C). The data suggest that duck endothelial cells react to HPAIV infection in a weaker pro-inflammatory and stronger antiviral fashion than chicken endothelial cells and duck fibroblasts.

### 3.5. Endothelial Cells Are a Target of HPAIV Infection in Duck Embryonated Eggs

To further study the susceptibility of duck endothelial cells to HPAIV infection, HPAIV tropism was assessed in duck embryos. We aimed to mimic the apical infection of endothelial cells following systemic dissemination of HPAIVs in vivo. This was achieved by inoculation of embryonated eggs in the allantoic cavity [74]. The allantoic route of inoculation results in replication of HPAIVs in the epithelial cell layers of the CAM. Subsequently, HPAIVs spread to the CAM vasculature and into the embryonic blood stream. ECE and EDE at intermediate (14-day-old ECE; 18-day-old EDE) and late (17-day-old ECE; 21-day-old EDE) gestational stages were inoculated with 10^3^ TCID_50_ of the GsGd-H5N1 HPAIV RG-A/turkey/Turkey/1/05. The embryo and CAM were harvested for further processing at 24 hpi (intermediate stage) or 48 hpi (late stage), upon which both species displayed ubiquitous subcutaneous hemorrhaging. Productive HPAIV infection was determined by immunohistochemical detection of NP. In both species, viral antigen was predominantly present in the epithelial cells of the CAM and in the endothelial cells of the CAM and embryonic tissues, including but not limited to the lungs, heart, liver, intestine, and kidney (Table 2). As an illustration, the endothelial cells of the lungs were positive for NP and showed evident histopathological changes in response to HPAIV infection (Figure 5). The developing avian lung contains parallel structures, the parabronchi, which consist of premature epithelial cells in tubular structures surrounded by mesenchyme (Figure 5A). The vasculature is embedded within the mesenchyme and can be recognized by the thin layer of endothelium which surrounds a lumen with nucleated erythrocytes. The lungs of infected ECE and EDE showed hyperemia and increased optical empty space in the mesenchyme, indicative of edema (Figure 5B). In both species, viral antigen was mostly confined to the endothelial cell layers. Taken together, the overt infection of endothelial cells in many embryonic duck tissues, including the respiratory tract, shows that embryonic duck endothelial cells are susceptible to HPAIV.

## 4. Discussion

Terrestrial poultry show a pronounced endotheliotropism upon infection with HPAIVs, which is associated with a plethora of systemic fatal clinical manifestations such as edema, hemorrhage, and coagulopathy. Contrastingly, HPAIVs do not, in general, infect endothelial cells of wild and domestic ducks, and viral antigen is predominantly detected in parenchymal and epithelial tissues instead [46]. Concomitantly, clinical manifestations upon HPAIV inoculation are generally milder. The reason for the species-dependent endotheliotropism has remained elusive partially due to the absence of robust endothelial cell culture methods of avian species other than chicken. We employed our previously established primary duck aortic endothelial cell cultures in combination with an in ovo setting to investigate the susceptibility of duck endothelial cells to HPAIVs. Here, we show that primary duck endothelial cells are susceptible to A/Vietnam/1203/04 and RG-A/turkey/Turkey/1/05, two H5N1 HPAIVs that cause severe disease in young Pekin ducks yet without endotheliotropism [75,76], upon direct inoculation in vitro and in ovo. However, primary duck endothelial cells showed a markedly different innate immune response than primary chicken endothelial cells, which was associated with reduced viral replication.

While the cellular distribution of sialic acid moieties has been extensively studied in mammals, information on SA moiety expression in avian tissues is limited, especially regarding endothelial cells. Kuchipudi et al. reported that adult chicken kidney endothelium expresses solely α2,6-linked SA moieties, showing a striking absence of α2,3-linked SA moieties [77]. A similar expression pattern was observed in embryonic chicken lung endothelium [74]. Contrastingly, primary chAEC in this study and previously published work [54] were detected by MAL-II lectin, which is specific for α2,3-linked SA moieties. However, in light of the limited information available on chicken endothelial cell-specific SA expression and contradicting results from FFPE-tissue lectin binding studies in general [77,78], a definitive conclusion regarding SA expression in chicken endothelial cells in vivo cannot be drawn. The results from the current study are in accordance with the high susceptibility of chicken endothelial cells in vivo to HPAIVs that have a preference for α2,3-linked SA moieties. Primary dAEC expressed both α2,6- and α2,3-linked SA moieties, as was reported for endothelial cells in adult mallard and Pekin ducks [77,79].

The absence of viral antigen in endothelial cells of mallard and domestic ducks upon HPAIV inoculation via the natural route has led to the suggestion that duck endothelial cells are refractory to HPAIV infection [46]. Although we detected a trend towards lower initial HPAIV infection percentages in dAEC, both in the present study using the strain A/Vietnam/1203/04 and previously using RG-A/turkey/Turkey/1/05 [56], the susceptibility of primary dAEC to HPAIVs suggests that duck endothelial cells are not inherently resistant to direct infection by HPAIVs. Additionally, dAEC sustained HPAIV multi-cycle replication, alluding to the presence of furin-like proteases. Endothelial cells were widely infected in duck embryos when those were inoculated with HPAIV via the allantoic route. Consequently, the absence of endotheliotropism of HPAIVs in wild and domestic ducks in vivo might be attributed to other factors, cellular or soluble, that are not accounted for in the current experimental system. For example, Short et al. have shown that human endothelial cells were not infected by H5N1 HPAIV when co-cultured with human epithelial cells, despite being susceptible to HPAIV infection in monoculture [80]. Epithelial cells are the first target of HPAIVs and might act as barrier that counteracts viral dissemination into the cardiovascular system or that indirectly alters the antiviral state of endothelial cells through paracrine signaling. Only one study, to our knowledge, has performed an in vivo investigation into the cellular tropism of HPAIVs (GsGd-H5N1) upon intravenous inoculation in domestic ducks, eliminating the prior targeting of epithelial cells [81]. No or minimal numbers of NP-positive endothelial cells were detected, indicating the resistance of duck endothelial cells to HPAIVs, even when apically targeted in vivo. However, the ducks were dissected at different days after inoculation than the inoculated chickens, which prevents direct comparison of endotheliotropism between the species. Therefore, further research on endotheliotropism upon intravenous HPAIV inoculation in ducks is necessary to establish whether duck endothelial cells are intrinsically resistant to infection in vivo.

Here, embryonic aortic endothelial cells and embryonated eggs were used as model system to study HPAIV infections. Although embryonated eggs present an affordable and easily accessible source of material for primary cell isolation, their use is accompanied with some caveats. The immaturity of the embryonic endothelial cells, innate immune system, and structural features within the developing embryo might favor infection by HPAIVs. The innate immune system develops early during avian gestation, and 14-day-old ECE can mount a proper innate immune response [82]. Nevertheless, antiviral responses increase and mature during further development, which continues in the weeks post-hatch [83]. The infection of endothelial cells in duck embryos upon HPAIV inoculation via the allantoic route is in stark contrast with the previously mentioned data from intravenously inoculated adult ducks [81], which might allude to the immaturity of the embryonic tissues. The AEC in this study were harvested from late-stage embryos and cultured in growth factor rich medium that promotes endothelial cell differentiation. However, we cannot exclude that the differentiated AEC still possess an embryo-like phenotype. Additionally, endothelial cells are a heterogeneous population and are not equally targeted during HPAIV infection in chickens [16,19]. Thus, care should be taken before results obtained with one endothelial cell type are extrapolated to others. We previously compared the susceptibility to HPAIVs of duck aortic endothelial cells and endothelial progenitor cells from the bone marrow, which was comparable [56]. This suggests that our results are in part generalizable to endothelial cells from other sources. Once currently available protocols will be adapted to the isolation of endothelial cells from different tissues, the validation of our results in a wider range of avian endothelial cell types is desirable.

The innate immune responses during HPAIV infections in chickens and ducks have been studied in vitro and in vivo as a means to explain the stark differences in HPAIV susceptibility and pathogenesis between these species (reviewed in [71,72]). Care needs to be taken when results from these studies are compared and summarized as they contain a plethora of different HPAIV strains, experimental methods, sample times, and often describe only one of the two species. However, the pro-inflammatory cytokine induction is generally stronger in chickens than in ducks following HPAIV infection [40,65,73]. Furthermore, excessive cytokine responses have been observed following infection with some but not all HPAIV strains in terrestrial poultry [29,30,84], whereas they are less apparent in ducks [29,65]. Ducks are thought to mount a quick and robust antiviral response upon HPAIV infection, consisting of the overexpression of interferons, ISGs, and pattern-recognition-receptors (PRR) [65,85,86,87], but this is not always reported [40]. A similar induction of antiviral responses has been observed in chickens inoculated with HPAIV, which is remarkably often similar or stronger than in duck counterparts [61,73,88] and could be due to higher levels of viral replication. However, the chicken antiviral responses are unable to ameliorate or clear HPAIV infection. Differences in innate immune responses between chickens and ducks are often attributed to the absence of RIG-I in chickens [89], which is the main cytoplasmic RNA-sensing PRR, but whose absence is partially compensated by signaling through the MDA5 receptor [90,91,92]. Here, pro-inflammatory responses were indeed slightly stronger in chicken cells, as shown by a modest upregulation of pro-inflammatory chemoattractant cytokine *IL8* upon HPAIV inoculation in chAEC, as described before [54], whereas a decrease was observed in dAEC. Similarly, Tong et al. recently set out to understand the role of endothelial cells in cytokine induction in avian species [55]. In that study, they reported the upregulation of pro-inflammatory genes in chAEC upon HPAIV infection, whereas those genes were either not differentially expressed or downregulated in dAEC. The individual relative expression patterns for *IL6* and *IL8* do not align perfectly between our study and that of Tong et al. Downregulation of *IL8* was not detected in dAEC by Tong et al. and, in contrast to our results, the induction of *IL6* expression differed between chAEC and dAEC. Interestingly, Tong et al. showed that direct stimulation of the innate immune system by poly(I:C) treatment resulted in an even bigger difference in pro-inflammatory cytokine expression between chAEC and dAEC. In the present study, antiviral responses, indicated by *IFNB* and *RSAD2* expression, were stronger in dAEC than chAEC, which correlated with reduced HPAIV replication. Tong et al. did report the overall trend of a stronger antiviral response in dAEC compared to chAEC, as seen by the upregulation of ISGs such as *MX1* and *RSAD2*, but did not detect differences in relative expression levels of *IFNB* itself nor in HPAIV infection levels. The differences between our two studies might be explained by different experimental methods, viral strains, and cell preparations. Additionally, we observed a cell type specific effect of interferon and ISG induction in duck cells, as endothelial cells showed stronger responses than embryonic fibroblasts. Based on the current results, we postulate that duck endothelial cells are more potent than other duck and chicken cell types in mounting antiviral responses, which might explain reduced and absent virus replication in vitro and in vivo, respectively. To allow for generalization of the current results in duck endothelial cells to HPAIVs in general, the panel of tested virus strains requires extension as only two viral strains were used to generate the current data. Moreover, further studies are warranted to clarify what other factors might limit the replication of HPAIV in duck endothelial cells and how endotheliotropism influences the immunopathology that accompanies HPAIV infections.

## 5. Conclusions

The present study provides insight into the susceptibility and innate immune responses of duck endothelial cells to HPAIV infection. Although dAEC were permissive to HPAIV infection, multi-cycle virus replication in dAEC was limited when compared to chAEC. Moreover, dAEC displayed a markedly different innate immune response than chAEC. These differences may contribute to the species-dependent endotheliotropism in vivo and consequently HPAIV pathogenesis in chickens and ducks.

## Figures and Tables

**Figure 1 viruses-14-00165-f001:**
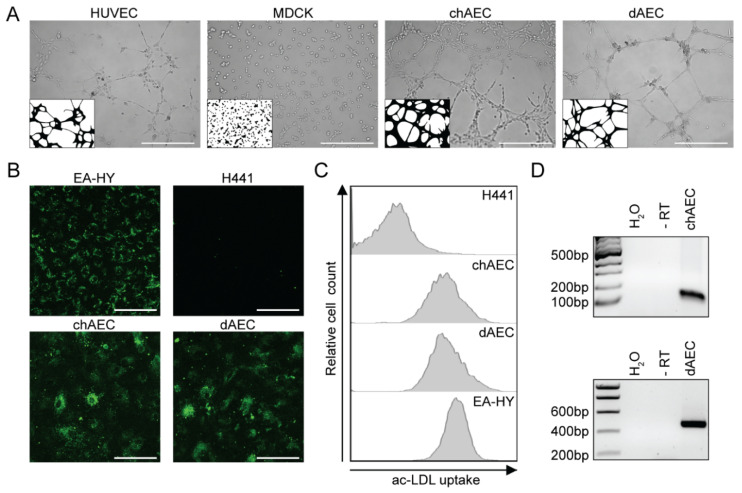
Endothelial cell characteristics of chicken and duck aortic endothelial cells. ChAEC and dAEC were isolated and passaged as described in the Materials and Methods section. (**A**) Representative bright field microscopy pictures of a tube formation assay after 4 h performed with chAEC (p14) and dAEC (p15). Human endothelial cells (HUVEC, p10) and canine epithelial cells (MDCK, p54) were used as positive and negative controls, respectively. In each corner, a schematic representation of the formed structures is shown. The scale bar represents 500 µm. Representative immunofluorescence images (**B**) and flow cytometry histograms (**C**) showing uptake of Alexa Fluor^®^488-conjugated ac-LDL by chAEC (p15) and dAEC (p16) after 4 h of treatment. Human endothelial cells (EA-hy, p17) and human epithelial cells (H441, p19) were used as positive and negative controls, respectively. The scale bar represents 100 µm. (**D**) RT-PCR for vWF expression on chAEC (p16) and dAEC (p17). The left lane is the DNA size marker; bp = base pairs. ‘-RT’ = samples where RNA was used as template as a control for genomic DNA detection.

**Figure 2 viruses-14-00165-f002:**
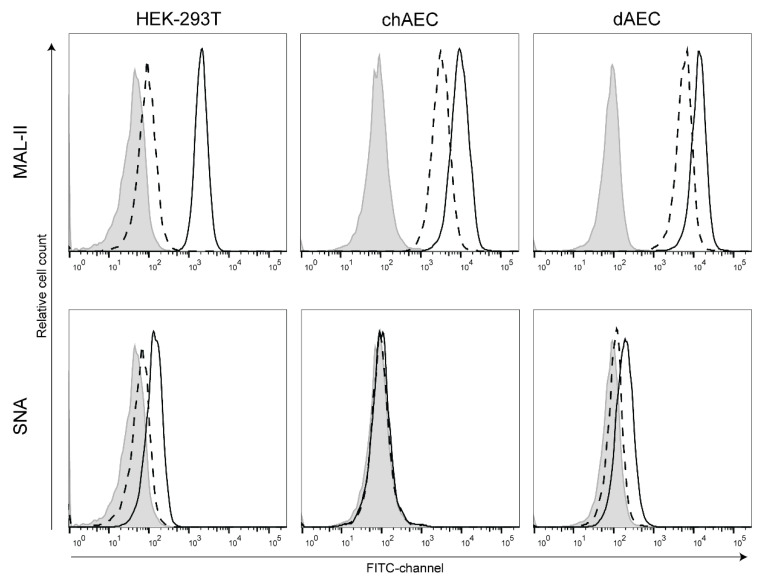
Sialic acid expression on the surface of primary avian endothelial cells. HEK-293T cells (p29), chAEC (p15), and dAEC (p16) were incubated for 2 h in medium with (dashed line) or without (black solid line) sialidase. Cells were stained with MAL-II (α2,3-specific) or SNA (α2,6-specific) lectins and subjected to flow cytometry. The negative control (in grey) consisted of untreated cells from which the lectins were omitted during the staining process. Representative flow cytometry histograms of at least three independent experiments are shown.

**Figure 3 viruses-14-00165-f003:**
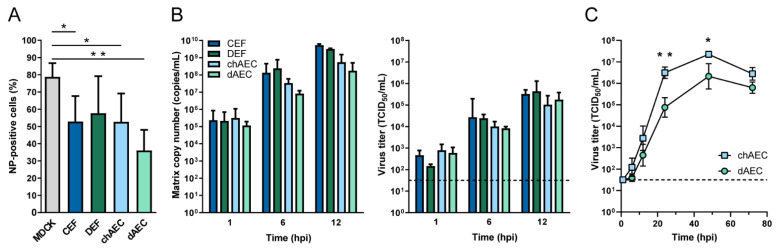
Replication kinetics of H5N1 HPAIV in duck and chicken endothelial cells. (**A**) MDCK, CEF, DEF, chAEC, and dAEC were inoculated with A/Vietnam/1203/04 H5N1 virus isolate at an MOI of 1. Cells were harvested at 6 hpi, and infection percentages were determined by NP staining and flow cytometry. Bars indicate the mean of four biological replicates, and the error bars represent the SD. (**B**) CEF, DEF, chAEC, and dAEC were inoculated as described for panel A, and supernatants were harvested at the indicated time points. Viral copy numbers were quantified by RT-qPCR on the matrix gene segment (left), and infectious virus titers were determined by endpoint titration in MDCK cells and expressed as TCID_50_/mL (right). Bars indicate the mean of three biological replicates, and the error bars represent the SD. Dotted line represents the limit of detection of the endpoint titration assay. (**C**) ChAEC and dAEC were inoculated with A/Vietnam/1203/04 H5N1 virus isolate at an MOI of 0.001. Supernatants were harvested at the indicated time points, and infectious virus titers were determined by endpoint titration in MDCK and expressed as TCID_50_/mL. Data are presented as mean ± SD from three independent experiments. Dotted line represents the limit of detection of the endpoint titration assay. Statistically significant differences were determined by one-way ANOVA followed up with individual unpaired t-tests. * *p* < 0.05; ** *p* < 0.01.

**Figure 4 viruses-14-00165-f004:**
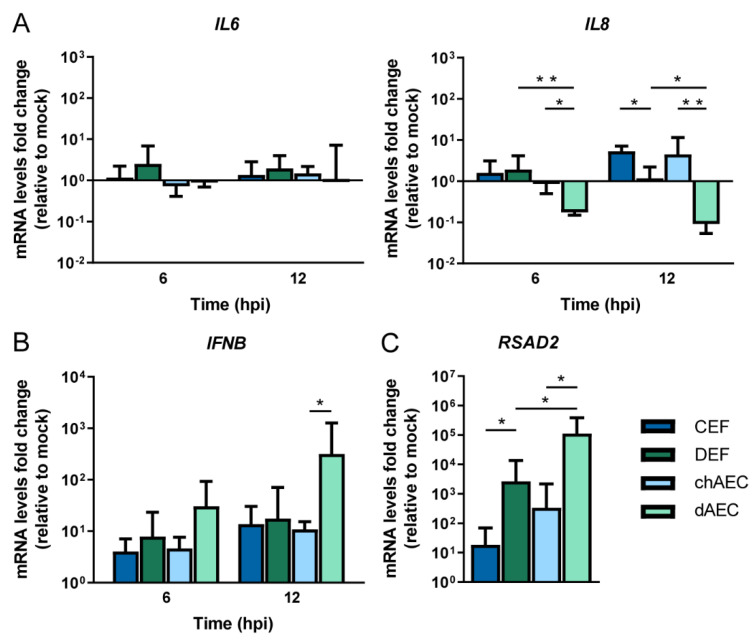
Host innate immune responses of avian primary cells upon HPAIV H5N1 inoculation. Monolayers of CEF, DEF, chAEC, and dAEC were inoculated with A/Vietnam/1203/04 H5N1 virus isolate at an MOI of 1. The cells were harvested at 6 and 12 hpi and analyzed for gene expression differences as compared to mock-inoculated cells. Messenger RNA levels were determined by a nucleic acid dye-based (**A**,**C**) or primer/probe (**B**) RT-qPCR. Fold changes were calculated using the 2^−ΔΔCT^ method with *GAPDH* serving as a housekeeping gene for normalization. (**C**) contains data from the 12 hpi time point. Bars indicate three biological replicates, and the error bars represent the SD. Statistically significant differences were determined on log-transformed fold changes by one-way ANOVA, followed up with individual unpaired t-tests. Only intra-species or intra-cell type significances are depicted. * *p* < 0.05; ** *p* < 0.01.

**Figure 5 viruses-14-00165-f005:**
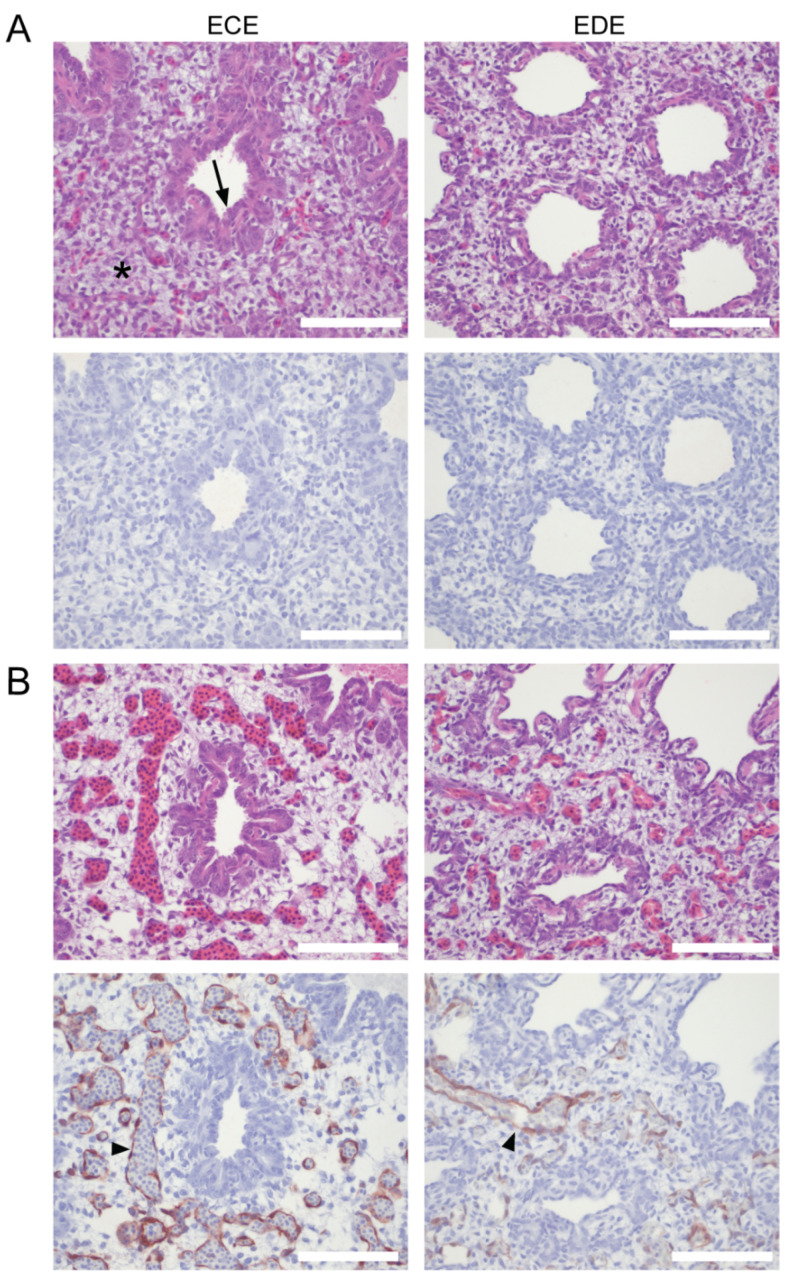
Cell tropism of H5N1 HPAIV in lungs of chicken and duck embryonated eggs. Fourteen-day-old ECE and 18-day-old EDE were mock-inoculated with PBS (**A**) or inoculated with 10^3^ TCID_50_ RG-A/turkey/Turkey/1/05 H5N1 (**B**) via the allantoic route (two embryos per condition). Embryos were harvested at 24 hpi and processed for immunohistochemistry. Top panels show HE staining. Bottom panels show influenza A virus NP staining as indicated by a red precipitate. Representative images of the lungs are shown. The mesenchyme (asterisk), epithelial cells from the premature parabronchi (arrow), and NP-positive endothelial cells (arrow head) are indicated. The scale bar represents 100 µm.

**Table 1 viruses-14-00165-t001:** Target genes and primers and probes sequences used during RT-(q)PCR.

(Nucleic Acid Dye-Based) RT-(q)PCR ^a^					
Target	Host	Accession Number	Oligo	Sequence (5′ to 3′)	Reference
*GAPDH*	Chicken	AF047874	Forward	GGTGCTAAGCGTGTTATCATCTCA	[60]
			Reverse	CATGGTTGACACCCATCACAA	
	Duck	XM_027449739.1	Forward	GCCTCTTGCACCACCAACT	This study
			Reverse	GGCATGGACAGTGGTCATAA	
*IL6*	Chicken	HM179640.1	Forward	GCGAGAACAGCATGGAGATG	[61]
			Reverse	GTAGGTCTGAAAGGCGAACAG	
	Duck	AB191038	Forward	GCAACGACGATAAGGCAGATG	[62]
			Reverse	TCTTATCCGATTTCAGCTTTGTGA	
*IL8*	Chicken	NM_205498.1	Forward	CTGCGGTGCCAGTGCATTAG	[54]
			Reverse	AGCACACCTCTCTTCCATCC	
	Duck	NM_205498	Forward	AGGACAACAGAGAGGTGTGCTTG	[63]
			Reverse	GCCTTTACGATCCGCTGTACC	
*RSAD2*	Chicken	NM_001318443.1	Forward	GCCGAGATTATGCTGTTGCTT	This study
			Reverse	TGATTAGGCACTGGAACACCT	
	Duck	KP198582.1	Forward	GCCGAGAGTATGCTGTTGCTT	[64]
			Reverse	AATGAGCAGGCACTGGAACAC	
*VWF*	Chicken	BK007988.1	Forward	GTGCTAAAACCTGCCAGAGC	This study
			Reverse	CACAACGTTCACCATCAAGG	
	Duck	XM_005012640.3	Forward	ACCACATGTTAGTGAGGAAC	[56]
			Reverse	CTTGGTAGGGTATGCTTCTC	
**Primer/probe RT-qPCR**					
*GAPDH*	Chicken/Duck	Ch: NM_204305.1 Du: XM_027449739.1	Forward	AGTCGGAGTCAACGGATTTG	This study
			Reverse	GGCYACCACTTGGACTTTGC	
			Probe	6-FAM CTGGTGACCAGGCGSCCAATAC BHQ1	
*IFNB*	Chicken	NM_001024836.1	N.A. ^b^	Gg03344129_s1 (Thermo Fisher Scientific)	N.A.
	Duck	KM035791.2	Forward	CTTTTGGACACCGACAAC	[65]
			Reverse	AGGATGTTGAAGAGGTGTTG	
			Probe	6-FAM CAAGCCGCACACACCGCC BHQ1	
Matrix gene segment	Influenza A virus	N.A.	Forward	CTTCTRACCGAGGTCGAAACGTA	This study
			Reverse	TCTTGTCTTTAGCCAYTCCATGAG	
			Probe	6-FAM TCAGGCCCCCTCAAAGCCGARA BHQ1	

^a^ RT-(q)PCR = Reverse Transcriptase-(quantitative) Polymerase Chain Reaction. ^b^ N.A. = not applicable.

**Table 2 viruses-14-00165-t002:** CAM and embryonic tissues of ECE and EDE containing NP-positive endothelial cells upon inoculation with HPAIV H5N1.

Species	Age at Inoculation (Days)	Age at Harvest (Days)	CAM ^a^	Lungs	Heart	Liver	Intestine	Kidney
Chicken	14	15	+/+ ^b^	+/+	+/+	+/+	+/+	+/+
Chicken	17	19	+/+	+/+	+/+	+/† ^c^	+/+	N.D. ^d^
Duck	18	19	+/+	+/+	+/+	+/+	+/+	+/+
Duck	21	23	+/+	+/+	+/+	+/†	+/+	N.D.

^a^ CAM = chorio-allantoic membrane. ^b^ The presence of NP-positive endothelial cells is shown with a + and was determined in two embryos per condition. ^c^ † = infection of endothelial cells could not be unambiguously determined; the absence of NP-positive endothelial cells combined with abundant necrotic endothelial cells is suggestive of prior viral infection. ^d^ N.D. = not determined.

## Data Availability

Data are contained within the article or Appendix A.

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
