# Peer review of "Reduced Replication of Highly Pathogenic Avian Influenza Virus in Duck Endothelial Cells Compared to Chicken Endothelial Cells Is Associated with Stronger Antiviral Responses"

_viruses, 2022, doi:10.3390/v14010165_

Round 1

Reviewer 1 Report

This study has found the association between the reduced replication of HPAIV and the stronger antiviral responses in duck endothelial cells by inoculating HPAVI in primary chicken and duck aortic endothelial cells for measuring virus replication kinetics and mRNA levels of pro-inflammatory cytokines. It is important to elucidate the mechanism of the species-specific endotheliotropism of HPAIVs. The authors presented well-designed experiments, clear results, and comprehensive discussions. However, the experimental design of this study is very close to reference 55. It would be more insightful if the authors can compare more transcriptional data by RNAseq from reference 55 to this study. One minor suggestion is that the first sentence in the section 2.1.”Cell culture” would be more clear if less information is put together or the sentence is reorganized.

Author Response

We thank the reviewers for their constructive critiques. Please find their comments reproduced below, followed up by each of our answers in a green font.

Comments and Suggestions for Authors: Reviewer 1

This study has found the association between the reduced replication of HPAIV and the stronger antiviral responses in duck endothelial cells by inoculating HPAVI in primary chicken and duck aortic endothelial cells for measuring virus replication kinetics and mRNA levels of pro-inflammatory cytokines. It is important to elucidate the mechanism of the species-specific endotheliotropism of HPAIVs. The authors presented well-designed experiments, clear results, and comprehensive discussions.

We thank the reviewer for their kind words regarding the experimental design, obtained data, and conclusions from our article.

However, the experimental design of this study is very close to reference 55. It would be more insightful if the authors can compare more transcriptional data by RNAseq from reference 55 to this study.

Thanks for this comment. The researchers from reference 55 entitled “Primary Chicken and Duck Endothelial Cells Display a Differential Response to Infection with Highly Pathogenic Avian Influenza Virus” utilized comparable chicken and duck aortic endothelial cell cultures for analyzing innate immune responses following HPAIV H5N1 inoculation at high multiplicity of infection, but using different strains. Our manuscript expands upon this subject by including the kinetics of HPAIV replication in endothelial cells, the addition of embryonic fibroblasts to discern species-specific differences from cell type-specific differences, different strains, and in ovo models. Additionally, the aims of the two manuscripts differ as our research focuses on the absence of endotheliotropism in ducks whereas reference 55 aims to elucidate the role of endothelial cells in cytokine storm induction in avian species. We agree with the reviewer that the experimental design regarding host response analyses in reference 55 and our manuscript is similar and the conclusions regarding pro-inflammatory responses are comparable. However, there are differences in individual gene expression levels, especially regarding the antiviral response. As requested by both Reviewer 1 and 3, these have been elaborated upon in the discussion section of the manuscript (lines 534-550).

One minor suggestion is that the first sentence in the section 2.1.”Cell culture” would be more clear if less information is put together or the sentence is reorganized.

This issue has been addressed accordingly to improve readability and clarity.

Reviewer 2 Report

In this manuscript, de Bruin et al. determine infection and replication kinetics of HPAIV in chicken and duck aortic endothelial. Additionally, they determine mRNA levels of several host innate immune genes. The experiments and presentation of the data are written clearly, and the conclusions are appropriate for the author’s findings. Although it would be interesting to see if any of the immune response findings affect HPAIV infection/replication, this reviewer understands that the time it takes to optimize those assays (including finding appropriate reagents since the authors are not working with human or mouse cells) can take a long time and are appropriate for a separate manuscript/the current manuscript can stand on its own. No changes are requested.

Author Response

Comments and Suggestions for Authors: Reviewer 2

In this manuscript, de Bruin et al. determine infection and replication kinetics of HPAIV in chicken and duck aortic endothelial. Additionally, they determine mRNA levels of several host innate immune genes. The experiments and presentation of the data are written clearly, and the conclusions are appropriate for the author’s findings. Although it would be interesting to see if any of the immune response findings affect HPAIV infection/replication, this reviewer understands that the time it takes to optimize those assays (including finding appropriate reagents since the authors are not working with human or mouse cells) can take a long time and are appropriate for a separate manuscript/the current manuscript can stand on its own. No changes are requested.

We thank the reviewer for understanding the additional challenges that are inherent to working with unconventional animal species in a laboratory setting. We are indeed continuing this work to confirm the hypotheses that were proposed in the discussion and conclusion sections of the current manuscript and hope to be able to share those results with the scientific community when finalized.

Reviewer 3 Report

Reviewer:

The article entitled “Compared to Chicken Endothelial cells is Associated with Stronger Antiviral Responses Reduced Replication of Highly Pathogenic Avian Influenza Virus in Duck Endothelial Cells” by de Bruin and colleagues mainly investigated the endotheliotropism of HPAIVs in primary chicken and duck aortic endothelial cells. This is a straightforward paper and their findings revealed that H5N1 HPAIVs are susceptible in primary duck endothelial cells but showed significantly lower viral replication and different innate immune responses compared to chicken endothelial cells. Based on the findings, they assumed that the different innate immune responses in dAEC contributes to the species-dependent differences in endotheliotropism and consequently HPAIV pathogenesis in ducks.

The experiments in this manuscript are well designed and some advancement of their previous publication (Davis et al., 2018; BMC Microbiology), but there are some issues that should be addressed.

Comments:

Line 131: It seems that the authors have used PR8 reassortant virus for A/ turkey/Turkey/1/05 (H5N1). The authors should briefly mention this in the methods section and replace A/turkey/Turkey/1/05 with RG-A/turkey/Turkey/1/05 throughout the manuscript (i.e line numbers 243; 397; 424).

The authors have used Wilt-type HPAIV (A/Vietnam/1203/04) strain for in vitro experiments whereas high-growth and low pathogenic PR8 backbone of H5N1 (A/turkey/Turkey/1/05) was used for the in ova/in vivo studies. The difference in the pathogenicity or infectivity of the two different backbone viruses (WT and PR8-backbone) makes it difficult to compare the results from the two different experiments. To avoid inconsistency in the experimental results, the authors should include at least one type of virus in all the experiments.

Recently, similar study by Tong et al.,2021 (Genes, 12, 901“Primary Chicken and Duck Endothelial Cells Display a Differential Response to Infection with Highly Pathogenic Avian Influenza Virus”) also demonstrated that chicken and duck endothelial cells display a different transcriptional response to HPAI H5N1 infection in vitro but chickens displaying a more pro-inflammatory response to infection. It would be useful if the authors could discuss more about this and compare with their findings in the discussion section.

Author Response

Comments and Suggestions for Authors: Reviewer 3

The article entitled “Reduced Replication of Highly Pathogenic Avian Influenza Virus in Duck Endothelial Cells Compared to Chicken Endothelial cells is Associated with Stronger Antiviral Responses” by de Bruin and colleagues mainly investigated the endotheliotropism of HPAIVs in primary chicken and duck aortic endothelial cells. This is a straightforward paper and their findings revealed that H5N1 HPAIVs are susceptible in primary duck endothelial cells but showed significantly lower viral replication and different innate immune responses compared to chicken endothelial cells. Based on the findings, they assumed that the different innate immune responses in dAEC contributes to the species-dependent differences in endotheliotropism and consequently HPAIV pathogenesis in ducks.

The experiments in this manuscript are well designed and some advancement of their previous publication (Davis et al., 2018; BMC Microbiology), but there are some issues that should be addressed.

We thank the reviewer for the positive reception of the manuscript.

Comments:

Line 131: It seems that the authors have used PR8 reassortant virus for A/ turkey/Turkey/1/05 (H5N1). The authors should briefly mention this in the methods section and replace A/turkey/Turkey/1/05 with RG-A/turkey/Turkey/1/05 throughout the manuscript (i.e line numbers 243; 397; 424).

Thanks for this comment. Our apologies for not having been clear enough in the material and methods section, but the A/turkey/Turkey/1/05 virus is not a PR8 reassortant. This recombinant HPAIV is a wild type virus as it contains all 8 segments of A/turkey/Turkey/1/05 and was produced through reverse genetics. To avoid confusion, we have clarified this in the revised manuscript (lines 133-134). We agree with the reviewer that RG should be added to the strain name and implemented this throughout the manuscript.

The authors have used Wilt-type HPAIV (A/Vietnam/1203/04) strain for in vitro experiments whereas high-growth and low pathogenic PR8 backbone of H5N1 (A/turkey/Turkey/1/05) was used for the in ova/in vivo studies. The difference in the pathogenicity or infectivity of the two different backbone viruses (WT and PR8-backbone) makes it difficult to compare the results from the two different experiments. To avoid inconsistency in the experimental results, the authors should include at least one type of virus in all the experiments.

We thank the reviewer for these comments. As clarified in the section above, the A/turkey/Turkey/1/05 virus is also wild type, albeit rescued through reverse genetics rather than being an isolate. Therefore, the two viruses used in this study are both wild type H5N1 HPAIVs that can cause severe disease in young Pekin ducks, but without endotheliotropism (PMID: 19023759 and PMID: 17494561). In a previous publication, we have shown that duck endothelial cells are susceptible to A/turkey/Turkey/1/05, but that they are infected to a lesser extent than chicken endothelial cells (PMID: 30340527). This corresponds with the results obtained using A/Vietnam/1203/04 strain in the current manuscript. To clarify this point, text was added in the discussion (lines 443-446 and lines 465-467). Furthermore, the results from the in vitro and in ovo experiments are not inconsistent as both show the intrinsic susceptibility of (embryonic) duck endothelial cells to H5N1 HPAIVs. We think that repetition of the experiments is not essential for the scope of the current article, as both viruses represent the same type of virus.

Recently, similar study by Tong et al.,2021 (Genes, 12, 901“Primary Chicken and Duck Endothelial Cells Display a Differential Response to Infection with Highly Pathogenic Avian Influenza Virus”) also demonstrated that chicken and duck endothelial cells display a different transcriptional response to HPAI H5N1 infection in vitro but chickens displaying a more pro-inflammatory response to infection. It would be useful if the authors could discuss more about this and compare with their findings in the discussion section.

We thank the reviewer for this suggestion. Indeed, reference 55 entitled “Primary Chicken and Duck Endothelial Cells Display a Differential Response to Infection with Highly Pathogenic Avian Influenza Virus” utilized chicken and duck aortic endothelial cell cultures, which resulted in partially overlapping conclusions regarding pro-inflammatory responses. As requested by both Reviewer 1 and 3, these have been addressed in the discussion section of the manuscript, as there are individual gene expression differences, especially regarding the antiviral responses (lines 534-550).

Round 2

Reviewer 3 Report

The manuscript by de Bruin and co-authors has undergone significant improvement and responded most of my comments.